# Nanomechanical and Morphological AFM Mapping of Normal Tissues and Tumors on Live Brain Slices Using Specially Designed Embedding Matrix and Laser-Shaped Cantilevers

**DOI:** 10.3390/biomedicines10071742

**Published:** 2022-07-19

**Authors:** Vladislav M. Farniev, Mikhail E. Shmelev, Nikita A. Shved, Valeriia S. Gulaia, Arthur R. Biktimirov, Alexey Y. Zhizhchenko, Aleksandr A. Kuchmizhak, Vadim V. Kumeiko

**Affiliations:** 1Institute of Life Science and Biomedicine, Medical Center, School of Biomedicine, Far Eastern Federal University, 690922 Vladivostok, Russia; frear.far@mail.ru (V.M.F.); shmelev.m.e@gmail.com (M.E.S.); nikitawayfarer@yandex.ru (N.A.S.); gulaya.lera@gmail.com (V.S.G.); biktimirov.ar@dvfu.ru (A.R.B.); zhizhchenko@iacp.dvo.ru (A.Y.Z.); alex.iacp.dvo@mail.ru (A.A.K.); 2A.V. Zhirmunsky National Scientific Center of Marine Biology, Far Eastern Branch of Russian Academy of Science (FEB RAS), 690041 Vladivostok, Russia; 3Institute of Automation and Control Processes, Far Eastern Branch of Russian Academy of Science (FEB RAS), 690041 Vladivostok, Russia

**Keywords:** atomic force microscopy, vibratome, nervous, neural, cell and tissue mechanics, glioma, glioblastoma, neuroblastoma, meningioma, diagnostics

## Abstract

Cell and tissue nanomechanics has been intriguingly introduced into biomedical research, not only complementing traditional immunophenotyping and molecular analysis, but also bringing unexpected new insights for clinical diagnostics and bioengineering. However, despite the progress in the study of individual cells in culture by atomic force microscopy (AFM), its application for mapping live tissues has a number of technical limitations. Here, we elaborate a new technique to study live slices of normal brain tissue and tumors by combining morphological and nanomechanical AFM mapping in high throughput scanning mode, in contrast to the typically utilized force spectroscopy mode based on single-point probe application. This became possible due to the combined use of an appropriate embedding matrix for vibratomy and originally modified AFM probes. The embedding matrix composition was carefully developed by regulating the amounts of agar and collagen I to reach optimal viscoelastic properties for obtaining high-quality live slices that meet AFM requirements. AFM tips were rounded by irradiating them with focused nanosecond laser pulses, while the resulting tip morphology was verified by scanning electron microscopy. Live slices preparation and AFM investigation take only 55 min and could be combined with a vital cell tracer analysis or immunostaining, thus making it promising for biomedical research and clinical diagnostics.

## 1. Introduction

The viscoelastic properties of the cell microenvironment have a significant impact on cell fate and behavior, affecting adhesion, migration, differentiation and proliferation [1]. Cells have special cytoskeleton-based feedback loops to maximize their mechanosensitivity [2]. Matrix synthesizing cells can influence the fate of stem cells and regulate the properties of differentiated cells through the mechanoreceptors. The mechanism is especially important for the tissues of the central nervous system, since their extracellular matrix (ECM) possesses a highly specialized structure. The predominance of the carbohydrate component over the protein one resulting in low stiffness can be attributed to the important properties of the brain matrix, thus directing the cell fate by limiting the mobility and proliferation of differentiated neural cells, while maintaining neurite outgrowth [3]. Therefore, it is obvious that in carcinogenesis, the ECM hardly has a minor role than neurons and glial cells. Moreover, experimental evidence reports that the viscoelastic characteristics and the microarchitectonics of the matrix determine the state of cancer cells and can serve as diagnostic criteria for different types of gliomas [4]. The nanomechanical properties of the matrix produced by glioma-associated stem cells determine the activity of the migration and metastasis, as well as the tumor growth [5]. Migrating tumor cells undergo significant molecular and cellular transformations through changing the level of cell–cell and cell–matrix adhesion and rearranging the cytoskeleton [6,7]. Moreover, in the course of chemo- and radiation therapy, the ECM can undergo certain changes, reflected in the accumulation of the fibrous component aggravating the disease [8]. Not only the matrix, but also individual tumor cells differ in their viscoelastic properties. The structure of these differences is much more complex than “one cell is tougher than the other’’, as tissue mechanics is influenced by appearance of actin stress fibers [9], production of matrix enzymes (e.g., metalloproteinases 2 and 9) [10], expression of vimentin, GFAP, microtubules [11] and effects of energy metabolism changes [12]. Therefore, live tissue should be studied as an integral object, considering cells together with their matrix [9,13]. Moreover, by analyzing the morphology in combination with viscoelastic properties, we can link the mechanics to certain objects–cells, receptors, matrix molecules, etc. In addition, the reason of the increase in the tumor tissue rigidity has not yet been precisely established [14]. Moreover, tumor cells bearing different sets of driver mutations differ markedly from the normal cells in terms of viscoelastic properties [10,15]. Thus, tissue mechanics investigation using atomic force microscopy became widespread technique in cancer research [16,17,18].

However, currently, AFM-based methods for live tissue analysis suffer from a number of limitations. The most important of these are low resolution or small size of investigating areas. Tissues with comparatively high stiffness, such as skin or cartilages, can be easily measured using AFM [19,20]. However, in the study of nervous tissue, which is very soft, mechanical and morphological mapping is completely absent in several research works [21,22]; in some of them, the resolution is measured in terms of tens of microns [23]. A few works present topographic maps of the nervous tissue at the submicron level, but they lack nanomechanical mapping [24], while the correlation of microarchitectonics and nanomechanics is considered the greatest interest.

Nowadays, AFM of live biological objects is commonly focused on investigating the cell lines grown in the cell culture dishes [25]. When it comes to tissue level, morphology is examined on previously fixed samples [26] or cryo-sections [27], precluding the nanomechanical mapping, as the stiffness of fixed and live tissue is markedly different [28]. When live tissue is studied by AFM, no morphology is obtained due to difficulties with uneven and torn surfaces [29]. Viscoelastic properties are usually investigated by macro methods such as rheometry [30]. Mechanical properties, being highly variable among cells and ECM at the microscale [31], are measured by the micro-rheometer method, which lacks morphological characterization. Thus, the rheological properties of live tissues are not directly linked to their micromorphology and structure [19]; this stems from the AFM technical limitations.

This work provides a new technique of brain sample preparation using a vibratome in combination with a special embedding matrix and high throughput scanning of live slices by AFM applying originally modified probes. The new technique allows high-resolution nanomechanical mapping associated with live cells and ECM native microarchitectonics analysis on a tissue level in a relatively short time, avoiding classical histological fixing and slicing artefacts.

## 2. Materials and Methods

### 2.1. Tissue Sample Preparation

The normal brain tissue samples were collected from the cerebral cortex of the Wistar rats (220 ± 20 g, 8–12 weeks old). Animals were anesthetized with 50 mg/kg sodium pentobarbital (Sanofi, Hannover, Germany). The skull was trepanned with a scalpel, then further opened along the anatomical sutures with large surgical scissors. The vault of the skull was broken off with tweezers, and the nerve plexuses were cut off from the brain with a surgical spatula; the brain was taken out of the cranium with tweezers and placed in a vessel with 10 mL of Neurobasal Plus Medium (Gibco, New York, NY, USA), which was put on ice. The large hemispheres of the brain were cut off with scalpel and placed in an embedding matrix, preliminarily formed in the bottom cone of a 50 mL Falcon type tube (Eppendorf, Hamburg, Germany), which was used as a mold.

Surgically resected glioma samples were obtained only from patients who signed the informed consent form for study participation. The use of human tissue material and animal experiments involving rats were approved by the FEFU Ethics Committee according to Resolution #5/19 December 2017.

Resected human tumor material was placed in the vessel containing Neurobasal Plus Medium and kept on ice. The sample was studied under a stereomicroscope (Stemi 508, Carl Zeiss, Jena, Germany) for the presence of capillaries and vessels, as well as zones of necrosis. An integral fragment of the brain, not less than 5 mm in length, was placed in an embedding matrix, preliminarily formed in the bottom cone of a 50 mL Falcon type tube, which was used as a mold.

### 2.2. Embedding Matrix

Hydrogels of various compositions were created by adjusting the ratios of agar and type I collagen (Sigma-Aldrich, St. Louis, MO, USA). Afterwards, these hydrogels were used as embedding matrices for obtaining live slices of tumor tissues and normal brain. A total of twenty variants of such gels were tested by measuring the rheological properties (stiffness and elasticity) using an atomic force microscope (BioScope Resolve, Bruker, Billerica, MA, USA) in Force volume mode (tip vertical movement amplitude: 1500 nm; ScanRate: 0.3 Hz; force setpoint: 1 nN) to make indentation deepth of AFM tip smaller than the probe diameter to meet the parameters of the Hertzian model [32]. In addition, the morphology of several gels was examined in PeakForce Quantitative Nanomechanical Mapping mode of 128 × 128 pixels.

The embedding matrix consisted of three components: agar, collagen I, and a gelation initiator. The stock solution of collagen I was prepared on acetic acid. The gelation initiator contained the required amount of NaOH to neutralize acetic acid and HEPES-NaOH (Rigaku Reagents, Tokyo, Japan); pH 7.4 used as the main buffer pair to maintain physiological pH. A series of hydrogel samples were obtained using 150 mM NaCl, 24 mM HEPES-NaOH, pH 7.4, 1.3 mM acetic acid, 2.1 mM NaOH as constant components, and 0.5 to 2.5% agar, 0.5 to 1.5 mg/mL collagen I as variable components.

Immediately before placing the tissue in an embedding matrix, the agar was melted in microwave oven at +90 °C, cooled down and kept in the water bath at (EcRos, Moscow, Russia) +37 °C. The embedding gel was composed by strongly mixing 1 mL of a gelation initiator and 1 mL of type I collagen of various stock concentrations (from 2.5 to 7.5 mg/mL). After mixing, 3 mL of agar of various stock concentrations (from 0.8 to 4.2%) was added and the resulting composition was transferred to the cone-shaped mold (bottom part of a 50 mL Falcon tube). The tissue samples were placed in the composed embedding gel just before the solidification and then transferred to be on ice for 5 min.

### 2.3. Vibratome Sectioning of Brain Samples

The cones with embedded samples were removed from the molds after solidification (Figure 1a), and glued onto glass slides (Menzel-Glaser, Thermo, Waltham, MA, USA) using cyanoacrylate glue (Weiss, Lauterach, Germany). The glass slide was fixed to the vibratome tray as shown in Figure 1b.

Samples embedded in the matrix were sliced using the vibratome EasiSlicer (Ted Pella, Inc., Redding, CA, USA) with the following parameters: the minimum blade feed rate (0.2 mm/sec) and the maximum vibration frequency (50 Hz); the angle of inclination of the blade to the tissue was adjusted to 13° (Figure 1b) (Appendix A: normal speed and normal amplitude). Slices ranging from 50 to 500 μm in thickness were produced and transferred using a weighing spatula into a Petri dish with 3 mL of Neurobasal Plus Medium.

A drop of cyanoacrylate glue was applied to a dry Petri dish; then, just before the solidification, the glue was removed with a pipette forming a thin film of glue on the surface of the dish for attaching a slice to it. DNA was released from dead cells and formed an adhesive layer on the slice was removed by 2 mL of DNase solution (1000 Units/mL DNase I, RNase-free, Thermo Fisher, Waltham, MA, USA) added for 10 min; afterwards, the immobilized slice was put in the 10 mL of Neurobasal Plus Medium.

### 2.4. Modification and Evaluation of AFM Probes

Two types of modified probes were originally prepared and evaluated.

We prepared the probes with a large tip radius by attaching a 10 μm diameter polystyrene microbead to the end of the tipless cantilever NP-10 (Bruker, Billerica, MA, USA) using cyanoacrylate glue.

Moreover, we modified the specific commercially available AFM probes (Scan Assyst High Resolution, Bruker, Billerica, MA, USA) with the cantilever profile, which decreases the hydrodynamic forces while scanning. The tip radius of the cantilever was increased via thermal reshaping under direct multi-pulse laser irradiation with sub-nanosecond laser pulses (a central wavelength of 533 nm, a pulse duration of 0.5 ns, a pulse repetition rate of 1.5 KHz, and an average power of 20 μW) generated by a passively Q-switched laser (PULSELAS, ALPHALAS, Gottingen, Germany). To laterally confine the melting region, the laser radiation was focused on the cantilever tip with a high numerical aperture (NA) objective of 0.9, which provides a full width at half the maximum of the focused beam of about 0.4 μm. Moreover, the laser intensity was additionally localized near the tip, owing to the high refractive index of Si and its high absorbance at 532 nm, as well as the truncated geometry of the cantilever [26]. These features allowed us to control the tip geometry of the cantilever and prevent its thermal deformation.

As-supplied and modified AFM probes were visualized using a scanning electron microscope (SEM, Sigma, Carl Zeiss, Germany) at an acceleration voltage of 20 kV.

### 2.5. Atomic Force Microscopy

AFM investigation was performed using Bruker BioScope Resolve (Bruker, Billerica, MA, USA). Agar-collagen gels applied for tissue embedding were analyzed in Force Volume semi-contact mode. This mode implies low interaction of the probe with the sample, which allows scanning without the risk of damaging the sample. Ten samples of each gel were analyzed in seven regions of interest.

The tissue micromorphology and nanomechanics were obtained in PeakForce Quantitative Nanomechanical mapping mode on live slices of the rat brain and human glioblastoma samples. Ten samples of all types were analyzed. Each sample was analyzed in seven regions of interest obtaining 128 × 128 pixels maps.

We tried several types of probes to compare the quality of scanning including the probe modified by attaching a 10 μm diameter particle to an NP-10 cantilever, standard and laser-modified ScanAsyst-in-Fluid probes with spring constants of 0.06 N/m, 0.7 N/m and 0.7 N/m, respectively. The spring constant was determined by using thermal tuning as a non-destructive method for the calibration of AFM probes. Deflection sensitivity measurement was performed by Nanoscope touch calibration to convert the raw photodiode signal measured in Volts to the deflection in the cantilever expressed in nanometers.

The spherical tip shape of the AFM probe and low indentation forces allowed us to use the Hertzian indentation model. We adjusted the scanning parameters in accordance with the scanning area and sample roughness. The indentation force was limited to make indentation depth smaller than tip radius to provide the possibility of nanomechanical property calculation. The fitting formula is presented below:(1)$$\left(F^{\frac{2}{3}}\right)=\left(\frac{4}{3}\frac{E}{\left(1-\nu^{2}\right)}\right)\sqrt{R}{)}^{\frac{2}{3}}\delta$$

This is the linearized Herz equation of interaction between a spherical indentor and elastic surface, where

*F* = force (from force curve);

*E* = Young’s modulus (fit parameter);

*ν* = Poisson’s ratio (sample dependent, typically 0.2–0.5);

*R* = radius of the indenter (tip);

*δ* = indentation.

AFM scanning was executed at room temperature in HBSS medium (Gibco, New York, NY, USA).

NanoScope analysis 2.0 software (Bruker, Billerica, MA, USA) was used for nanomechanical analysis. The morphological polynomials were fitted by the computation of a single polynomial of a selectable order for an image and subtracts it from the image to remove the image artefacts.

### 2.6. Laser Scanning Microscopy of Live Slices Stained with FDA-PI Fluorescent Tracers

The tissue slices were stained for live/dead assay by adding the fluorescein diacetate (FDA) (Sigma-Aldrich, St. Louis, MO, USA) combined with propidium iodide (PI) (Sigma-Aldrich, St. Louis, MO, USA) and analyzed by laser scanning microscopy (LSM) using the confocal Z-stack mode. The FDA was added to a final concentration of 8 μg/mLand PI to 50 μg/mL, incubated for 20 min in the dark at room temperature. Series of optical tissue sections (Z-stacks) were captured in two channels (FITC and TexasRed) using the upright optical module of a laser scanning microscope FV1200 (Olympus, Tokyo, Japan) with a 25× water-immersion contact lens. Two-dimensional LSM projections of a series of optical sections projected onto a plane were obtained using the FV10-ASW 4.1 software. The surface of the obtained sections was reconstructed by a transparent-to-opaque transformation using the Imaris 7.6.5 software package.

### 2.7. Immunohistochemical Analysis Combined with AFM

After the AFM investigation, rat brain live slices were fixed with 4% paraformaldehyde prepared on PBS (Gibco, New York, NY, USA) for 12 h at 4 °C, washed 5 times with PBS containing 0.05% Tween 20, with cycles running 10 min each. The indirect immunohistochemical analysis was provided using primary rabbit polyclonal antibody against rat glial fibrillary acidic protein (Anti-GFAP Antibody, Polyclonal, 60128, Stemcell Technologies, Vancouver, BC, Canada) and secondary antibody conjugated with fluorescent dye against rabbit immunoglobulins (Goat anti-Rabbit IgG (H+L) Highly Cross-Adsorbed Secondary Antibody, Alexa Fluor 488, A11034, Thermo Fisher Scientific, Waltham, MA, USA) according to the manufacturer’s protocol. The slices were counterstained with 1 μg/mL 4′,6-diamidino-2-phenylindole dihydrochloride (DAPI, Thermo Fisher Scientific, Waltham, MA, USA) for 5 min, washed 3 times with PBS and analyzed using AFM and LSM as stated above.

## 3. Results

### 3.1. Obtaining Live Slices of Brain Specimens Embedded in the Matrix

To obtain the optimal embedding matrix, we tested a panel of matrices with different agar to collagen I ratios. The data on the stiffness and elasticity measured for all the tested agar and collagen I concentrations are displayed in Table 1.

Agar gels without collagen did not allow us to obtain slices suitable for AFM, because their elasticity and plasticity were low; even 2.0 and 2.5% agar gels were characterized by low elasticity (less than 24.9 kPa), combined with relatively high stiffness (up to 3.6 µN/m). Figure 1f demonstrates that the vibratome blade left striations on a 2.5% agar-only gel surface. However, 2.5% agar gel supplemented with 0.5 mg/mL type I collagen was suitable for obtaining high-quality slices, avoiding striation artefacts (Figure 2g).

To pick up an appropriate embedding matrix formulation, gels with different agar-collagen I concentrations were produced and their viscoelastic properties—together with their suitability for vibratomy and AFM scanning—were analyzed. Several agar gels with a collagen supplement were unsuitable for obtaining slices, because a certain elasticity and stiffness were required. Gels with an agar concentration of up to 1.5% combined with a collagen I concentration of 0.5–1.0 mg/mL were inappropriate for obtaining slices due to their low elasticity, as a very soft matrix collapsed and could not hold the tissue (Figure 2c). Collagen I supplemented at a concentration of 1.5 mg/mL to 1.5% agar gel greatly improved the vibratomy results and allowed high-quality slices to be obtained. Extremely stiff gels resulted from 2.5% agar gel and collagen I in all investigated concentrations. An excessively stiff matrix squeezed the sample during cutting (Figure 2e).

The appropriate values of elasticity and stiffness were from 26.7 kPa to 28.7 kPa and from 2.4 µN/m to 3.38 µN/m, respectively, which matched the 1.5% agar gel with 1.5 mg/mL of collagen and the 2.0% agar gel with a collagen concentration from 0.5 mg/mL to 1.5 mg/mL These formulations provided the best quality of vibratomy (Figure 2d), as the slice surface did not show strong relief changes more than 20 μm in height, making it suitable for AFM. The correct angle of the blade was adjusted for tissue vibratomy. The angle of the blade to the tissue was set at 13°, which provided the best slice quality. We employed the gasket/spacer, because this blade angle was not supported by the vibratome model used in the study (Figure 2b). The vibratomy of nervous tissue in an appropriate embedding matrix has allowed us to obtain high-quality live tissue slices varying in thickness at least in the range of 50–1000 μm. The 500 μm thick slices were chosen for AFM investigation, because this thickness allowed the upper surface of the slice to stay alive and free of glue.

Next, we aimed to show that vibrosectioning in different matrices does not affect cell viability and the nanomechanical properties of slice. To do this, we obtained 10 sections in each of the matrices suitable for AFM requirements and compared their viability and rheological properties with a tissue fragment not subjected to vibratomy; the results are shown in Figure 1. Statistically, there are no differences in the mechanical properties and viability of the slices obtained in the different matrices and the tissue not subjected to vibratomy.

It is important to note that the parameters of vibratomy indicated in Section 2.3 are the only suitable ones. As the speed increases or the amplitude decreases, tissue sections are compressed or torn (Appendix A: high-speed, Appendix A: low-amplitude).

### 3.2. AFM Probe Characterisation

The AFM probes were first examined by SEM to visualize their tip profile prior to usage (Figure 3a,c,e). For comparison, the AFM scans of the surface of the 2% agar hydrogel were prepared using modified and as-supplied probes (Figure 3b,d,f). Figure 2a,b provides the SEM image of the stock ScanAsyst Fluid High Resolution probe (Figure 3a), as well as the high-resolution AFM image of the agar hydrogel surface obtained using this tip (Figure 3b), where the smallest gel pores with a diameter of up to to 50 nm and a depth of up to 200 nm can be easily resolved.

Figure 3c shows a similar cantilever, the tip of which was laser-reshaped to achieve a reduced tip curvature radius of 400 nm. The AFM visualization of the agar gel surface revealed the expectedly reduced resolution of the laser-modified cantilever (Figure 3d). Meanwhile, the 50 nm width surface pores still can be resolved, but with the maximal depth reduced to 20 nm, owing to the rounded tip geometry. Finally, the NP-10 cantilever with an attached microbead (Figure 3e) demonstrates an even lower resolution (Figure 3f), allowing us to visualize the surface pores larger than 100 nm.

### 3.3. Combined AFM and LSM Studies of Brain Live Slices

AFM probes equipped with the 10 μm diameter microbeads have already been used to study brain tissue nanomechanics, but without morphological mapping [33,34]. Therefore, we attested the similar microbead-based cantilever regarding its applicability for the nanomechanical and morphological mapping of tissues and compared its performance with the developed laser-modified probe. Of note, the as-supplied sharp tips were not suitable for sample scanning due to the large amount of image artefacts and bad quality of force curves.

According to the performed LSM studies (Figure 4a), the cell viability in the slices was high, as the number of live cells (green color produced by FDA bioconversion) significantly prevailed over the number of dead cells (red color of PI). The quantification of the tissue slices’ viability is shown in Figure 1. Neural cell bodies and processes can be clearly distinguished in the obtained image (Figure 4a). Two insets (Figure 4h,i) from Figure 4a—show the morphological maps of the rat brain slice surface reconstructed from regions of interest on the series of Z-stack images obtained by LSM in confocal mode. Additionally, these areas are marked in Figure 4a. Tuberous elongated formations can be discerned on the surface depicting the processes of neurons (neurites). Additionally, both reconstructed areas were chosen for morphological mapping (Figure 4b,c), nanomechanical mapping via elasticity (Figure 4d,f) and stiffness analysis (Figure 4e,g) that were performed with two types of the AFM cantilevers. As can be seen, the laser-modified cantilever allows us to clearly resolve the location and morphology of the processes (Figure 4b–e). The obtained AFM data are consistent with those obtained using LSM, with the exception of minor inaccuracies caused by errors in the AFM operation or associated with differences in the algorithms for constructing 3D structures in the NanoScope Analysis and Imaris software. The nanomechanical maps obtained by the laser-modified tip presented the cell processes as the formations with higher stiffness and elasticity compared to those for the ECM surrounding them (Figure 4e,f).

Figure 4c shows the surface morphology map and its 3D reconstruction obtained using the microbead-based tip. The shape and location of the cells reproduced structures similar to those visualized by LSM. However, the detailization of these structures was lower than those present in Figure 4b, which was due to their larger probe curvature radius. Figure 4f,g show nanomechanical maps of these surfaces indicating elasticity and stiffness, respectively. The differences in the viscoelastic properties of the cells and the ECM were less obvious due to the lower resolution of the probe used, which is proved by histogram.

Next, we aimed to explore the viscoelastic and morphological properties of human glioma samples using the characterization protocol applied above.

Cell viability was indicated by green fluorescence of FDA conversion, which was examined by an LSM investigation of 50 μm Z-stack projection, as shown in Figure 5a. Cells were closely packed and had round shapes that are characteristic of the proliferative zones of malignant tumors. The 3D reconstructions of two areas of the human glioma slice surface obtained via transparent-to-opaque LSM data transformation are shown in Figure 5h,i. These areas were further analyzed by laser-modified (Figure 5b,d,e) and microbead-based AFM cantilevers (Figure 5c,f,g) to perform morphological mapping (Figure 4b,c), nanomechanical mapping via elasticity (Figure 5d,f), and stiffness analysis (Figure 5e,g). Slice morphology obtained by AFM equipped with a laser-reshaped tip (Figure 5b) reliably reproduced the one reconstructed from LSM data (Figure 5h). The distinct heterogeneity in the viscoelastic properties (Figure 5d,e) of cellular and ECM structures was easily detected and corresponded to certain morphological characters (Figure 5b). The surface structure obtained by AFM with a microbead-based tip (Figure 5c) mostly corresponded to the LSM-based profile (Figure 5i), but was less detailed because the probe applied had too large of a tip radius and could not visualize deep narrow pores and cavities (Figure 5c). Generally, the nanomechanics data obtained with micro-bead based probe (Figure 5f,g) poorly correlated with tissue micromorphology on glioma slice, where cells and ECM could not be identified (Figure 5c).

Due to immunohistochemical analysis being a widespread technique in clinical laboratory diagnostics, a combination of AFM-based mapping with LSM acquisition of molecular marker distribution can become promising technique. Figure 6 shows that the method proposed allows the visualization of single cells on a slice surface and the distinction of their nucleus, body, and processes. Immunohistochemical staining is shown on Figure 6a, where cell nuclei were colored in blue with DAPI and the cytoskeleton of glial cells labelled in green with an antibody against GFAP. The AFM surface morphological map is shown on Figure 6c and almost ideally reproduces the LSM data. The cell nucleus and its processes were distinguishable; their shape and location coincide, to demonstrate that the LSM and AFM channels were superimposed on each other (Figure 6b). Figure 6 clearly demonstrates that the maximum resolution of atomic force microscopy is much higher than that of the laser one. Figure 6c is much more detailed than Figure 6a.

### 3.4. Quick Mapping and Characterization of Live Tissue Nanomechanics

Embedding the matrix composition developed for high-quality vibratomy of brain tissues, together with the techniques for probe preparation fitted to high throughput AFM-based nanomechanical mapping, were successfully applied to estimate the in situ viscoelastic properties of normal and malignant tissues of the central nervous system. The final method protocol is presented in Table 2. For the rat cerebral cortex, the elasticity and stiffness obtained by the probe with a 10 μm microbead attached were 0.413 ± 0.038 kPa and 0.56 ± 0.168 mN/m, respectively (Figure 4f,g). The AFM analysis of rat cerebral cortex using the ScanAsyst Fluid High Resolution laser-shaped probe showed that the elasticity and stiffness were 0.686 ± 0.15 KPa and 0.3 ± 0.08 mN/m, respectively (Figure 4d,e). For slices of human glioblastoma multiforme (G4), nanomechanical characteristics assessed with a 10 μm microbead-based probe were 0.842 ± 0.31 kPa for elasticity (Figure 5f) and 1.45 ± 0.536 mN/m for stiffness (Figure 5g). Correspondingly, the elasticity and stiffness of the same preparations alternatively measured with the laser-shaped tip were 1.292 ± 0.311 kPa and 0.8 ± 0.56 mN/m, which is shown on Figure 4d,e. These data are consistent with the measurements of live tissues of the central nervous system obtained by macroscopic rheological studies [34,35]. The laser-shaped probe allowed a more accurate measurement of stiffness and elasticity than the probe with an attached 10 μm particle.

## 4. Discussion

Atomic force microscopy has entered into biological sciences as a novel tool for cell and tissue research, supplementing different types of optical and electron microscopy. Combined nanomechanical and morphological AFM mapping provides a great opportunity in material sciences and industrial quality control. Our work is focused on a technique for studying nervous tissue, since it has the lowest stiffness, which makes its mapping difficult using AFM. For many other tissue types, significant advances have already been made in their analysis using atomic force microscopy [19,20]. The main problems arising in the analysis of brain live tissue have already been described in the works of scientists who pioneered the use of AFM. The first task is to fix the tissue sample for subsequent scanning. Most major professionals use a thin layer of biocompatible glue [29,36,37]. The second well-known problem is the high heterogeneity of the tissue and its uneven surface [29,37]. This problem was solved in one of the works by using AFM with a large z-range [29], although this method has several disadvantages, including the rendering high throughput mapping impossible. Another described method is the use of a probe with a large radius: up to 25 microns [36]. At the same time, increasing the probe radius helps to solve the problem of measuring the mechanical properties by averaging the result over a large area of the surface [36]. Previously, highly adhesive uneven surfaces with low elasticity and stiffness were analyzed only at microscale level. As for nanoscale measurements, the AFM assessment of the morphological features of normal and tumor tissues has been previously performed on fixed histological samples [26]. Chemical fixation significantly changes the morphological integrity and physico-chemical properties of the tissue. Another way to obtain morphological maps using AFM is based on cryo-sectioning [27], but this method has the same disadvantages as studying a chemically fixed tissue. Vibratome technique allows us to obtain a slice of the live tissue [38]. However, we have found that vibratome brain tissue slices are not suitable for AFM-based combined morphological and nanomechanical mapping, as their surface was uneven and torn. This is a well-known problem in general regarding postoperative tissues including brain samples [29].

We propose two main solutions for obtaining the combined maps of morphology and mechanics of live tissues using AFM, implementing a vibratomy with an embedding matrix and the use of the originally modified AFM probes.

Recently, the utility of the embedding medium has been proposed to improve the quality of slices [39,40]; because pre-fixed tissue [41] and infiltrating medium [40] were used, the advantages of a vibratomy were negated due to these procedures causing cell death. However, agarose as an embedding matrix can preserve the tissue alive, and is suitable for AFM investigation [36]. To provide AFM investigation in the mode of combined morphological and nanomechanical mapping, slices should be obtained in an embedding matrix with certain composition and viscoelastic properties, because they influence slice quality [39]. Taking into account these facts, we created an embedding matrix for live tissue vibratomy optimized for AFM investigation. This matrix, based on the collagen–agar hydrogel has several important properties. Firstly, it does not infiltrate the tissue, and therefore, preserves the slice’s nanomechanical properties. Secondly, it keeps the tissue sample vital throughout the study. Thirdly, the matrix rheological properties are optimized in order to obtain high-quality slices by protecting the tissue from tearing and crushing under the vibratome blade. Agar and collagen were chosen as the base components of the matrix because they are highly compatible with live objects and have been used for tissue embedding many times [36,42,43,44,45]. Agarose-only hydrogels have already been used as an embedding matrix for the vibratomy of live brain tissue [36], but they can be employed to obtain slices of live tissue suitable only for the Ramp Mode of AFM. However, as we have shown, such gels are not elastic enough (Table 1) to provide high quality slices for the PeakForce Quantitative Nanomechanical mapping mode of AFM. The striated hydrogel surface provided evidence of its insufficient elasticity (Figure 2f). It was found that the collagen I supplement provides the agar-based gel appropriate elasticity (Table 1), resulting in the absence of striations on the surface (Figure 5g). Different proteins have already been used for this purpose, including albumin [46] and gelatin. Gelatin has been reported as great improver of slice quality, making it flat enough for measurements in contact modes [47]. Recently, it was shown that an embedding matrix should have a certain composition [39]. We found that gels with an agar content ranging from 1.5 to 2%, with collagen supplement in concentrations from 0.5 mg/mL to 1.5 mg/mL, are characterized by having an elasticity ranging from 27 kPa to 28 kPa and stiffness from 2.5 µN/m to 3.1 µN/m (Table 1). After evaluating the ratio of live and dead cells in the sections obtained in different variants of the embedding matrix with certain composition and rheology makeups, we found that these properties of the matrix do not affect the viability of the slice, as well as its nanomechanical properties (Figure 1). Moreover, it has been established that there are no statistical differences in viability and rheology among a fragment of the nervous tissue not subjected to vibratomy and the sections obtained in various embedding matrices.

Using the collagen–agar hydrogels specified above, the combined morphological and nanomechanical mapping of normal rat brain and human glioma samples were successfully performed. The introduced approach was developed specifically for investigating a live tissue region of interest, with a size of 100 × 100 μm by means of AFM in the Force Volume Mode or even PeakForce Quantitative Nanomechanical Mapping. There are several similar techniques reported in the available literature, which suffer from several kinds of limitations, e.g., using a single measurement mode, exploring small areas, and missing simultaneous registration of morphology [21,22,23,24,29,33,37]. These limitations are caused by AFM’s inability to scan the irregular surfaces which have inequalities in height difference bigger than the Z-piezo element limitation.

The commercially available probes do not meet the requirements for tissue analysis on live slices. Firstly, the live tissue is excessively adhesive to the AFM probes. The originally modified probes used in this work do not have such a drawback, since one probe was modified with the laser, and the second probe was modified with a polystyrene microbead, both resulting in a decrease in adhesion. A large-radius probe was more suitable for the live tissue investigation, allowing us to avoid tip immersion in the tissue [33]. The ScanAsyst Fluid High Resolution probe has a characteristic sharp profile and a small tip radius, while the probe modified with the microbead has a larger radius (5 μm). The probe modified with the 10 μm microbead has already been used for the live tissue analysis [33]; however, these data miss the morphological investigation, presumably due to resolution issues that we found (Figure 3f). We managed to select the optimal dimensional characteristics of the probe for live tissue exploration by shaping it with a laser to a 400 nm tip radius (Figure 3c,d).

Additionally, the vibratomy technique improved in this study can be implemented in neurobiology and neurophysiology. Obtaining thin live sections of brain tissue and subsequent measurement of electrical potentials is not new; however, the use of the tip-integrated electrodes in AFM creates new frontiers in biomedical studies [48]. We suggest that the technique described here would be suitable for live slice analysis, bringing diversified research protocols to neuroscience by means of the possibility to combine physiological, morphological and nanomechanical investigation on the single slice. More frequently, brain tissue slices are analyzed by intraoperative immunohistochemical studies employing cryotomes for quick sectioning [27]. However, extremely low temperatures and water crystallization can damage the tissue and result in an insufficient quality of morphology visualized with antibody conjugates and counterstains. The technique proposed in this research allows the combining of immunohistochemical analysis with high-quality AFM mapping, which can be prospective in future clinical diagnostics (Figure 6). Correlating the morphology of slice surfaces obtained with AFM and LSM is an important part of the work. It should be clarified that, in general, the data of these studies are comparable (Figure 4, Figure 5 and Figure 6). Tissue structures detected with the help of LSM correspond to the AFM morphological maps. There are minor differences, which are explained by different operating principles of a laser microscope and contact force microscopy. In addition, the software used for the reconstruction of the three-dimensional structure has different algorithms. Otherwise, the morphological maps obtained using a laser-reshaped ScanAssyst Fluid High Resolution probe have enough resolution, which makes it possible to distinguish on the slice surface not only individual cells, but also their parts, such as nuclei or processes, which is shown in the Figure 4, Figure 5 and Figure 6. Morphological maps obtained with an NP-10 microbead-based probe have a significantly lower resolution; this can be seen in the Figure 4, Figure 5 and Figure 6. Thus, this paper proposes a useful technique for modifying stock probes for the needs of atomic force microscopy of live tissue.

The investigation of the mechanical features of the biological specimens is of great interest in biomedical research and clinical diagnostics, because different pathological mechanisms can modify the physico-chemical properties of tissues and cells. The micromechanical properties of live tissues can be measured using AFM, but in low-resolution mode and without morphology [49]. The macromechanical properties of live tissues have been already characterized and described in the available literature [50,51]. The main drawback of the rheological method is that it obtains an average value for an entire object, which does not describe the microscale variability; however, the average value can be implemented as an initial reference for high-resolution techniques. Viscoelastic properties of brain tissues were investigated using macroscopic measurements. According to rotational rheometry data, the elastic modulus for an intact rat brain ranges in some cases from 100 to 1200 Pa [52].

The nanomechanical data obtained in this study are consistent with the traditional rheometry and AFM in the Ramp mode, so the elasticity of the cerebral cortex of normal rat brain was 0.686 ± 0.15 kPa; moreover, the live slice analysis in the mapping mode allowed us to combine nanomechanics with morphology. This correlation prompts us to suggest that the high variability of the average elasticity values was caused by the difference in elastic properties of the ECM and nervous tissue cells (Figure 4b). This heterogeneity was resolved using a laser-modified probe with 400 nm tip radius. The 10 um microbead-based probe did not allow us to obtain differences between the cell and ECM, but the average elasticity was 0.613 ± 0.038 kPa, which is in the range mentioned above. Meanwhile, the elasticity of the glioblastoma (G4) is reported to vary in the range of 0.5–1.8 kPa, as measured by MR Elastography Analysis [50]. Glioma tissue slices were studied according to protocol developed in our research and average elasticity values were estimated at 1.692 ± 0.411 kPa, determined with the laser-modified tip, and 1.442 ± 0.310 kPa, using the 10 um microbead probe. Because our technique can distinguish cells from ECM, the cause for the change in the visco-elastic properties of the tissue can be established.

Recent studies showed changes in ECM composition under the radiation therapy of the glioblastoma multiforme, which is expressed by replacing carbohydrate components by fibrous ones with a protein content predomination [8]. The method of combined nanomechanical and morphological mapping described here is necessary to study the mechanism of this process and its quantitative characterization. This could be promising for future biomedical investigations and clinical diagnostics in the course of glioma treatment.

## 5. Conclusions

Until recently, methods of combined morphological and nanomechanical mapping of live tissues had serious shortcomings. The proposed methods were limited in their capabilities and suffered from several disadvantages. We managed to bridge this gap by proposing the use of a special embedding matrix for obtaining high-quality slices of live tissue and originally modified probes for high throughput mapping using AFM. The main advantages of the described technique for obtaining slices are its simplicity, the availability of reagents, and the short time required to obtain a sample. This makes it possible to study tissues in which the viability does not differ from the sample that is not subjected to vibratomy.

We elaborated a technique that firstly made it possible to study the nanomechanical properties of the tissue associated with its micromorphology. This is of great interest in investigating the mechanical heterogeneity of cells and their micro- and nanoachitectonics, which are characteristic of different cell populations in healthy and diseased central nervous systems. In addition, ECM components implicated in tissue pathogenic transformations typically resulted in stiffness increase in carcinogenesis. These tissue characteristics can be precisely analyzed by applying the technology described here. With appropriate refinement, it could be possible to use the main findings of this research for any other type of biological tissues. The technology for obtaining probes that meet the requirements for scanning live tissue is described, but it requires sophisticated laser equipment.

The combined study of molecular markers distribution in tumors by both laser and atomic force microscopy, obtaining certain nanomechanical and morphological patterns, may become extremely important and promising. This technique could be very promising for future clinical diagnostics, especially in combination with rapid immunohistochemical assay, which will allow the study of the molecular signatures of malignant brain tumors, the suitability of intraoperative assessments, and the development of personalized treatment strategies.

## Figures and Tables

**Figure 1 biomedicines-10-01742-f001:**
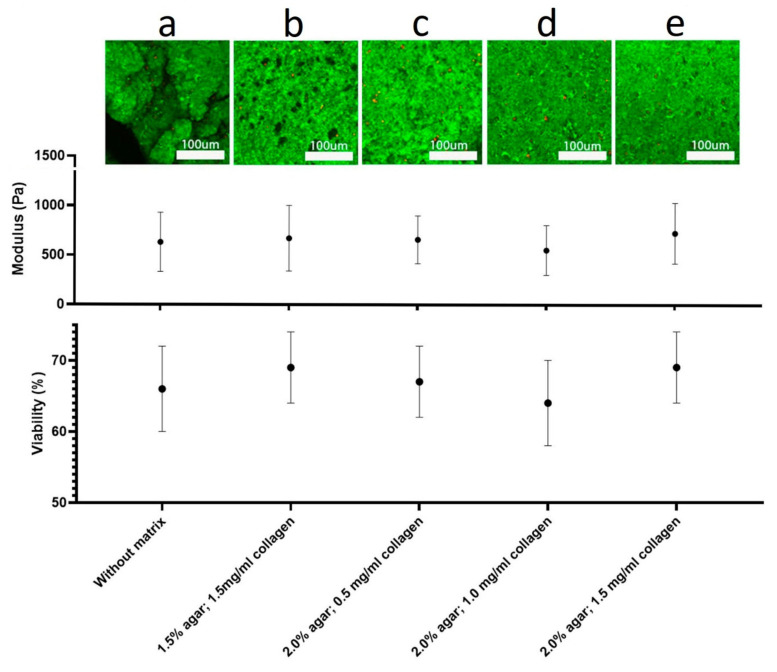
Comparative characteristics of the mechanics and viability of tissue sections obtained in different types of embedding matrix with tissue not subjected to vibratomy. (**a**) A tissue fragment not subjected to vibratomy in the matrix. (**b**) Tissue section obtained in a matrix consisting of 1.5% agar, 1.5 mg/mL collagen. (**c**) Tissue section obtained in a matrix consisting of 2.0% agar, 0.5 mg/mL collagen. (**d**) Tissue section obtained in a matrix consisting of 2.0% agar, 1.0 mg/mL collagen. (**e**) Tissue section obtained in a matrix consisting of 2.0% agar, 1.5 mg/mL collagen.

**Figure 2 biomedicines-10-01742-f002:**
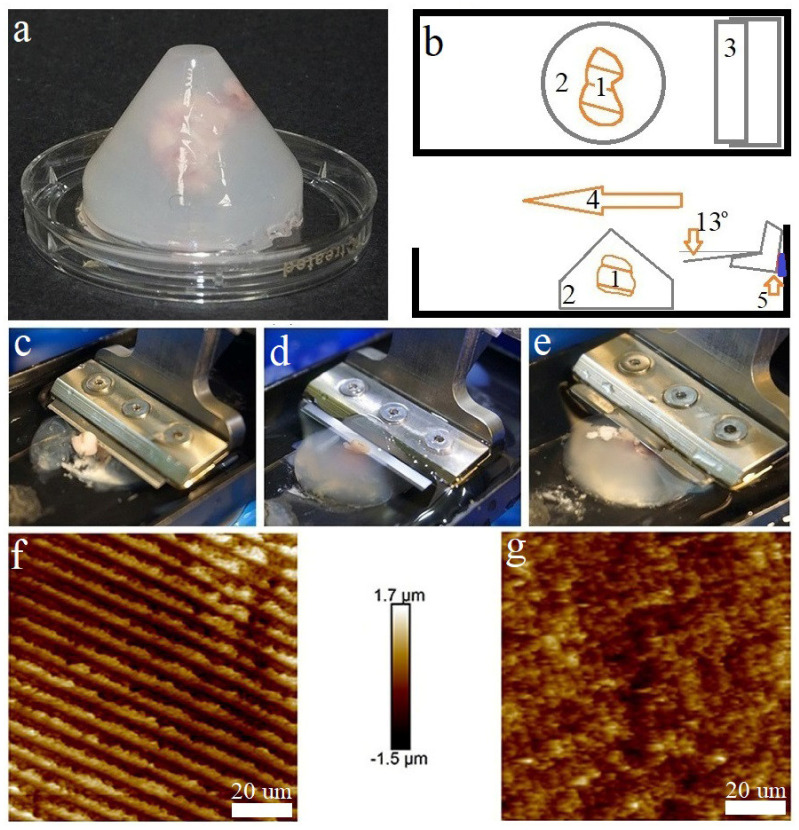
Vibrosectioning of the rat brain embedded in the matrix cone (**a**–**e**): (**a**) brain specimen embedded in the matrix cone; (**b**) schematic representation of the vibratome components adjusted for live brain sectioning: (1—specimen of brain, 2—embedding matrix cone, 3—vibratome blade, 4—vibratome blade translation, 5—gasket/spacer); (**c**) Vibrosectioning of the rat brain in the embedding matrix with insufficient stiffness (<2.500 µN/m) and elasticity (<27 kPa); (**d**) Vibrosectioning of the rat brain in the embedding matrix with suitable stiffness (2.6 µN/m) and elasticity (27 kPa); (**e**) Vibrosectioning of the rat brain in the embedding matrix with excessive stiffness (>3.5 µN/m) and elasticity (>30 kPa); (**f**,**g**) slice quality control by AFM with ScanAsyst Fluid High Resolution tip (Height sensor channel): (**f**) the slice surface of an agar gel (2.5% agar) without collagen (Height sensor channel); (**g**) the slice surface of collagen–agar gel (2.0% agar; 0.5 mg/mL type I collagen).

**Figure 3 biomedicines-10-01742-f003:**
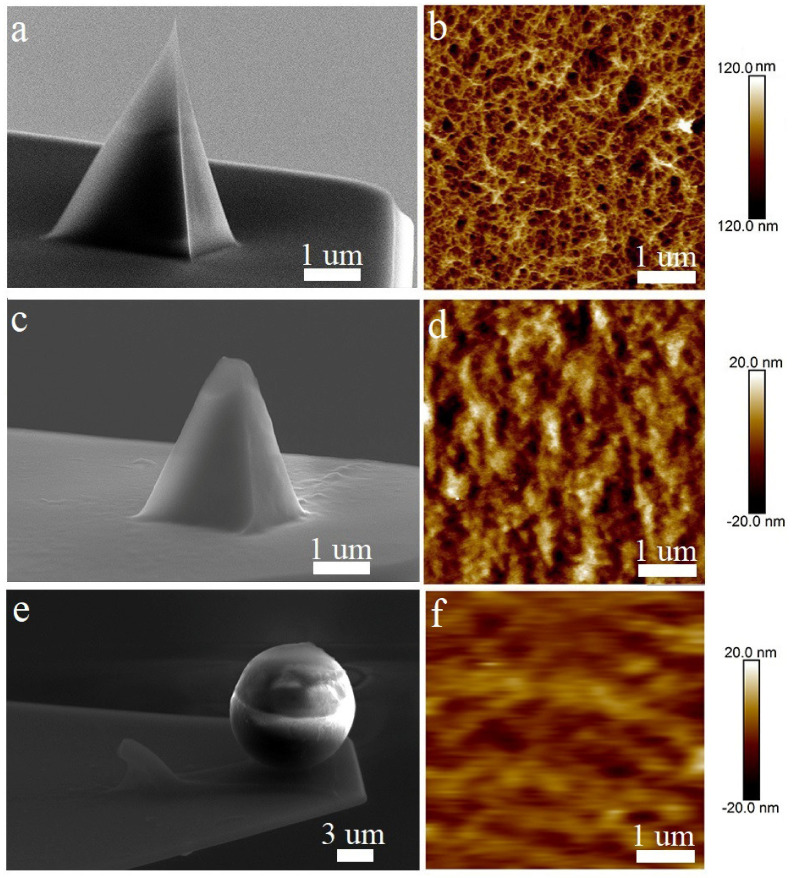
Comparative characteristics of the AFM probes: SEM images showing (**a**) stock ScanAsyst Fluid High Resolution probe and (**c**) laser-modified ScanAsyst Fluid High Resolution probes, as well as (**e**) NP−10 probe with an attached microbead. (**b**,**d**,**f**) AFM morphological maps of the 2% agar gel surface obtained using the above-mentioned cantilevers.

**Figure 4 biomedicines-10-01742-f004:**
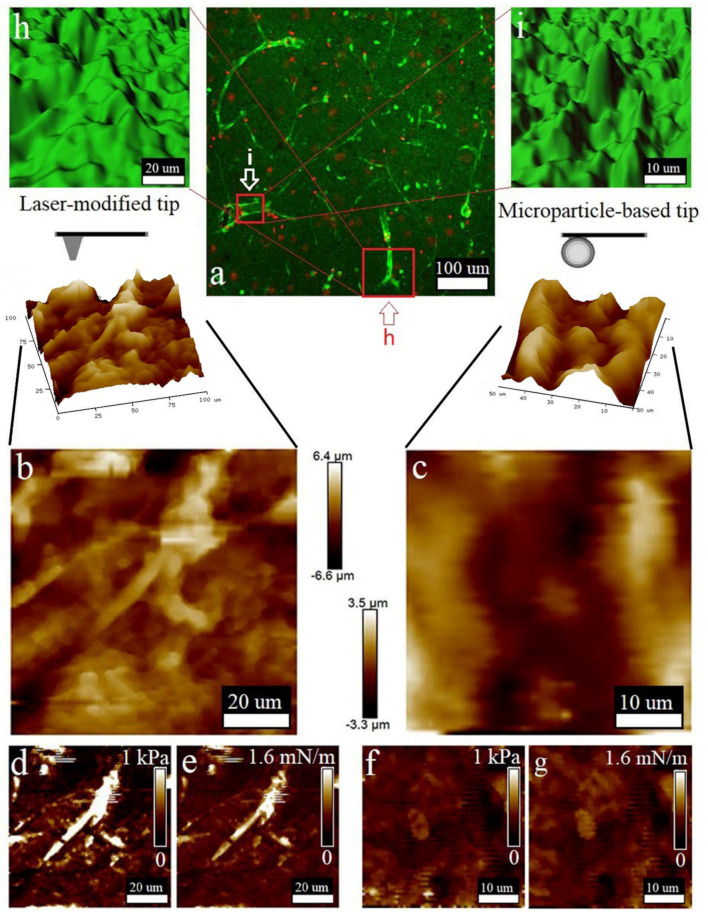
Surface properties of the rat brain slice studied by LSM (**a**,**h**,**i**) and AFM (laser-reshaped ScanAsyst Fluid High Resolution probe (**b**,**d**,**e**) and microbead-based cantilevers NP−10 (**c**,**f**,**g**). (**a**) Large−scale LSM map showing the ratio of live (green glow) to dead cells (red glow). (**h**,**i**) Two areas show 3D reconstruction of the smaller map sections chosen for subsequent AFM studies with two types of cantilevers. Surface morphology of the chosen slice obtained using laser-reshaped (**b**) and microbead-based (**c**) cantilevers. Insets on both images provide 3D reconstructions of these areas. (**d**,**e**) Elasticity and stiffness AFM maps by laser−reshaped ScanAsyst Fluid High Resolution probe. (**f**,**g**) Elasticity and stiffness AFM maps by microbead−based cantilevers NP−10.

**Figure 5 biomedicines-10-01742-f005:**
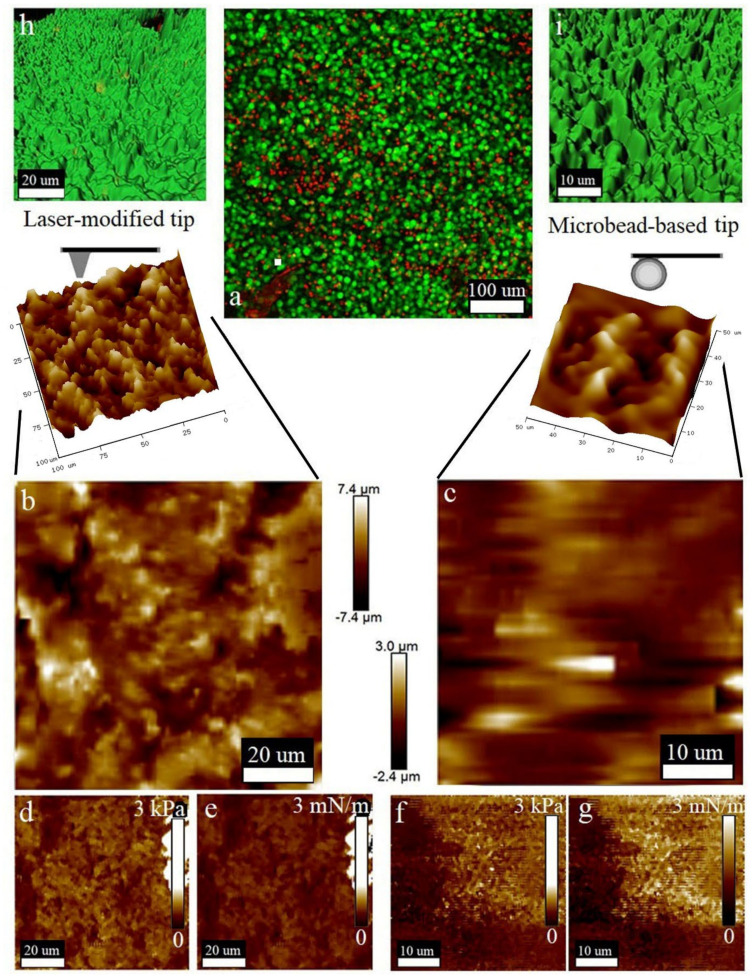
Surface properties of the human glioma slice studied by LSM (**a**,**h**,**i**) and AFM (laser-reshaped ScanAsyst Fluid High Resolution probe (**b**,**d**,**e**) and microbead−based cantilevers NP−10 (**c**,**f**,**g**). (**a**) Large-scale LSM map showing the ratio of live (green glow) to dead cells (red glow). (**h**,**i**) Two areas show 3D reconstruction of the smaller map sections chosen for subsequent AFM studies with two types of cantilevers. Surface morphology of the chosen slice obtained using laser-reshaped (**b**) and microbead−based (**c**) cantilevers. Insets on both images provide 3D reconstructions of these areas. (**d**,**e**) Elasticity and stiffness AFM maps of the chosen areas by laser-reshaped ScanAsyst Fluid High Resolution probe. (**f**,**g**). Elasticity and stiffness AFM maps by microbead-based cantilevers NP−10.

**Figure 6 biomedicines-10-01742-f006:**
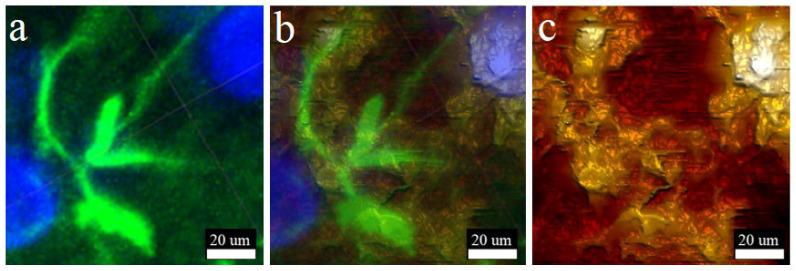
Combined immunohistochemical LSM and AFM analysis of rat brain slices. (**a**) Immunohistochemistry of slice surface captured by LSM (blue signal–DAPI-cell nucleus; green signal–GFAP-glial cell cytoskeletal protein); (**b**) merged LSM and AFM channels; (**c**) morphological map of the slice surface obtained by AFM (height sensor).

**Table 1 biomedicines-10-01742-t001:** Rheological/mechanical properties of different embedding matrix compositions.

Variable Components	Stiffness, µN/m	Elasticity, kPa
0.5% agar 0.5 mg/mL collagen	0.37 ± 0.03	02.83 ± 0.09
1.0% agar; 0.5 mg/mL collagen	0.8 ± 0.05	03.89 ± 0.22
0.5% agar; 1.0 mg/mL collagen	0.8 ± 0.05	04.32 ± 0.26
0.5% agar	0.64 ± 0.05	06.23 ± 0.27
0.5% agar; 1.5 mg/mL collagen	1.19 ± 0.07	07.63 ± 0.24
1.0% agar	1.05 ± 0.06	07.96 ± 0.36
1.5% agar	1.74 ± 0.09	09.38 ± 0.38
1.0% agar; 1.5 mg/mL collagen	1.68 ± 0.09	11.93 ± 0.32
1.0% agar; 1.0 mg/mL collagen	2.07 ± 0.1	13.97 ± 0.337
1.5% agar; 0.5 mg/mL collagen	2.26 ± 0.08	16.16 ± 0.35
1.5% agar; 1.0 mg/mL collagen	2.68 ± 0.09	20.86 ± 0.65
2.0% agar	2.69 ± 0.17	21.35 ± 0.67
2.5% agar	3.41 ± 0.18	24.15 ± 0.71
1.5% agar; 1.5 mg/mL collagen *	2.58 ± 0.1	27.62 ± 0.85
2.0% agar; 0.5 mg/mL collagen *	2.93 ± 0.1	27.46 ± 0.7
2.0% agar; 1.0 mg/mL collagen *	3.21 ± 0.17	27.62 ± 0.73
2.0% agar; 1.5 mg/mL collagen *	3.07 ± 0.18	27.82 ± 0.91
2.5% agar; 0.5 mg/mL collagen	3.73 ± 0.18	35.59 ± 0.94
2.5% agar; 1.0 mg/mL collagen	4.24 ± 0.26	46.38 ± 1.1
2.5% agar; 1.5 mg/mL collagen	4.35 ± 0.29	46.59 ± 0.27

* hydrogels provided high-quality slices.

**Table 2 biomedicines-10-01742-t002:** Protocol for live brain slices preparation and AFM mapping.

Action Needed	Timing
**Embedding matrix and mold preparation:** Prepare: 1. 2.5–3.3% agar on 200 мM NaCl solution (A) 2. 2.5–7.5 mg/mL Collagen I on 30 мM acetic acid (B) 3. Gelation initiator (150 mM NaCl, 120 mM HEPES-NaOH, pH 7.4, 1.0 мM NaOH (C) 4. Prepare the embedding mold by cutting off the conical bottom of the 50 mL Falcon tube	Prepare and keep in advance
**AFM tip preparation:** Equip your AFM instrument with a fluid imaging probe that has a tip radius of about 400 nm. The originally modified probe can be prepared by laser ablation using the standard ScanAsyst Fluid High Resolution Probe. The tip radius of the cantilever can be increased via thermal reshaping under direct multi-pulse laser irradiation with sub-nanosecond laser pulses (a central wavelength of 533 nm, a pulse duration of 0.5 ns, a pulse repetition rate of 1.5 KHz, and an average power of 20 μW) generated by a passively Q-switched laser. To laterally confine the melting region, the laser radiation should be focused on the cantilever tip with a high numerical aperture (NA) objective of 0.9, which provides a full width at half the maximum of the focused beam of about 0.4 μm.	Prepare and keep in advance
**Biopsy embedding:** Melt the agar (A) in the microwave and put it in water bath at a temperature of 37 °C for 5 min. Select an appropriate biopsy fragment under a stereomicroscope and transfer this with tweezers into the ice-cold DMEM. Put 3 mL of melted agar (A) in the embedding mold, add 1 mL of collagen I (B), add 1 mL of gelation initiator (C), and mix all the components by gentle aspiration. Immediately place the piece of tissue inside the mold, submerging it by the matrix. Place it in the center and avoid tissue contact with mold wall. Put the mold in ice bath for 5 min.	15 min
**Embedding cone****mounting:** Take the mold from ice bath. Turn the cone over and gently remove the cured embedding matrix. Stick the embedding matrix cone with cyanoacrylate glue onto a glass slide and fix it in the vibratome tray.	5 min
**Tissue vibrosectioning:** Attach the tray to the vibratome. Set the blade angle to 13°, the vibration frequency to 50Hz, and the movement speed to 0.2 mm/sec. Make several test sections with a thickness of 50 to 500 µm.	5 min
**Tiss****ue section mounting:** Apply a drop of cyanoacrylate glue on Petri dish using a pipette and immediately remove the glue. A thin film of glue should remain on the surface of the dish. Using tweezers, place the tissue section on the glue. Wait for 2 min and add DNase solution prepared on PBS (1000 U/mL) to the dish for 5 min at room temperature. Remove DNase; add ice-cold DMEM. The tissue section is ready for AFM analysis.	10 min
**AFM Scanning:** Calibrate the probe properly. Select the region of interest on the tissue section using an optical channel. Set it to PeakForce Quantitative Nanomechanical mapping mode and adjust the scan parameters to obtain a high-quality force curve.	~20 min (depends on resolution)
**Additional steps:** This may include some other procedures with tissue slices, already mapped by AFM: live cell tracing, live/dead assay, tissue fixation, and immunostaining.	depends on type of procedure

## Data Availability

Not applicable.

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
