# Peer review of "Nanomechanical and Morphological AFM Mapping of Normal Tissues and Tumors on Live Brain Slices Using Specially Designed Embedding Matrix and Laser-Shaped Cantilevers"

_biomedicines, 2022, doi:10.3390/biomedicines10071742_

Round 1

Author Response

Thank you very much for your thoughtful criticism. Your tips helped us to improve the quality of our manuscript.

Your recommendations were accepted by us. Below we have responded to each point of your comments.

  1. –Reviewer 1 wrote:

“The authors claim to measure live slides of tissue, but there is a tissue fixation for immunohystochemistry before AFM and LSM measurements, although in the protocol proposed in table 2 they do not perform the immunohistochemistry protocol and AFM is placed on alive tissue. This is confusing for the readers”.

Author’s response:

We want to clarify that immunotyping was done after the procedure of mapping live tissue by AFM. We have added the corresponding step to Table 2, placing it in the correct order and added clarification to text (Materials and methods, section 2.7, line 225). Thus, tissue fixation associated with cell death occurs after the live tissue has been mapped by atomic force microscopy.

  1. Reviewer 1 wrote:

“In figure 2, caption should be revised. It says “(a) 269 as-supplied and (b) laser-modified ScanAsyst Fluid High Resolution probes as well as (c) NP-10 270 probe with an attached microbead” and letters and images are not corresponding”.

Author’s response:

We have changed the caption for figure 3, 4, 5, considering your comments.

  1. Reviewer 1 wrote:

“In line 292-293 they reconstruct regions of interest based on the Z-stack images obtained by LSM. Why are they interesting related to the rest of tissue? It would be great if they depict or mark the neurites in the image.”

Author’s response:

In Figure 4a, we have added additional red frames that indicate which regions in Figure 4a correspond to Figures 4h and 4i. With the help of such a refinement, it is possible to understand which neurites were scanned using AFM and LSM. It should be clarified that the scanning angle may vary.

  1. Reviewer 1 wrote:

“In line 299, reference to the image is wrong. Instead fig. 3 b to e, it should be figure 3 b,d,e. It is also indicated that AFM data is consistent with LSM, but in figure 3 it is not clear if image h corresponds to reconstruction of image b, and image I with image c and this consistence is not clear. In line 332 they claim that AFM morphology map correspond to the one reconstructed by LSM. It could be interesting to add some marks and distinctives, as well as a scale bar in the LSM images to facilitate the visualization.”

Author’s response:

The visual correspondence of the AFM and LSM data is indeed not complete, however, we scanned the same tissue areas both with a laser scanning microscope and with an atomic force microscope. Moreover, the AFM probe was aimed at a tissue area using a fluorescent microscope, which made it possible to scan exactly the same area. Thus, all the conformity of LSM and AFM data declared by us are accurate and truthful. Differences in the three-dimensional structure of the same section of the slice surface are primarily associated with different methods of image acquisition, AFM contact mode, and fluorescence detection on the LSM. Fluorescence tracer stains mainly the cells, not ECM. The software algorithms for reconstructing a three-dimensional structure on different types of microscopes are also fundamentally different. In this case, it is fair to give preference to AFM data, since it directly interacts with the tissue surface. This is partly why we undertook such work, since such high-quality morphological maps of living nervous tissue had never been obtained with the help of AFM. We added clarification to manuscript (lines 539 – 553).

  1. Reviewer 1 wrote:

“In figure 5, a scale bar would be useful.’

Author’s response:

We have added a scale bar in Figure 5.

  1. Reviewer 1 wrote:

“Again, captions in figures 3 and 4 should be revised, as sometimes there is no concordance between the letter in the image and the letter mentioned in the caption.”

Author’s response:

Additionally, the captions for Figures 3, 4 and 5 have been clarified.

  1. Reviewer 1 wrote:

“As indicated in Methods, Young modulus and stiffness was calculated by DMT-model using PeakForce. In line 358, 361, 363 and 365, values of Young modulus and stiffness of rat brain and human glioma are given. (In line 361 it should say kPa instead of Pa?) As there is a huge heterogeneity in the tissues studied, how is this value obtained? Is it a mean of all the data obtained in each point of the measurement?”

Author’s response:

Quantitative data given for tissue elasticity are the mean and standard deviation. During the mapping of a tissue section at a resolution of 128 by 128 points, we obtained 16384 force curves for each tissue sample. Based on 16384 values of elasticity and stiffness, we determined the mean and standard deviation. This method was chosen in order to compare the obtained data with the results of macrorheological studies given in the literature, since most of the studies contain this format of data presentation. Indeed, the tissue is very heterogeneous in its nanomechanics, so we added the histograms of the distribution of elasticity in Figures 4 and 5. Protocol of AFM, including number of samples is described in lines 178-211.

  1. Reviewer 1 wrote:

“It would be interesting to compare disease and healthy tissue from the same animal model to explore the real potential of the described technique for diagnosis.”

Author’s response:

We did not aim to compare normal rat brain tissue and tumor human brain tissue with each other, we will try to take your comments into account in the following works. In this work, we compared the elasticity and stiffness of the normal rat brain tissue and human tumor obtained by our method with the literature data. Our data reliably correspond to the generally known ones (lines 566-580).

  1. Reviewer 1 wrote:

“In the introduction it is said that that there is a rearrangement of cytoskeleton due to molecular and cellular transformations, and this also change the mechanical properties of single cancer cells by different  mechanisms in each cell (lines 56-58). Please cite the relevant works:

  1. Alibert, C., Goud, B., & Manneville, J. B. (2017). Are cancer cells really softer than normal cells?. Biology of the Cell, 109(5), 167-189.

2.Yubero, M, et al. "Effects of energy metabolism on the mechanical properties of breast cancer cells." Communications Biology 3.1 (2020): 1-9”

Author’s response:

We studied the works you cited, found them interesting and useful for us, cited them in the introduction (lines 59-66).

Reviewer 2 Report

This paper describes a method to embed brain tissue samples for AFM imaging of morphological features. The authors argue that currently no method exists for living tissue AFM imaging. However, to our knowledge AFM living tissue imaging, even in vivo – both in terms of mechanical properties (i.e., modulus) and topography – has been performed multiple times. The advantages and the influence of the preparation method on the extracted tissue properties are not discussed or not quantitatively assessed. The methodology is not discussed in detail (e.g. AFM probe specifications and imaging parameters for all the discussed imaging modes are not given, also no info on the number of samples and data statistics). Cell viability is not quantified. Scale bars and imaging specs are not given for most of the figures. The language is sometimes confusing. The manuscript should be re-written.

See for instance

Krieg, M., Fläschner, G., Alsteens, D. et al. Atomic force microscopy-based mechanobiology. Nat Rev Phys 1, 41–57 (2019).  

https://www.ncbi.nlm.nih.gov/pmc/articles/PMC6587663/

The main novelty in respect to state of the art should be better discussed in the introduction session. This part should be re-written as it is unclear – some general statements are made without sufficient arguments to make a point. The language is sometimes confusing and should be improved. Specifically, the proposed technique should be introduced in the context of the literature relevant for tissue embedding for AFM to clarify novelty and advances in respect to state of the art. The choice of the embedding materials should be justified. Most relevant references should be cited.

Concerning the methodology  - section 2.5. AFM – all procedural details should be given – e.g. how many samples have been analyzed for each condition, fitting formula’s used for modulus and stiffness calculation.

Modulus maps are not clear in the manuscript – it would make sense maybe to include modulus histograms to quantify the effect of the different sample preparation conditions. It would be interesting to check the effect of the embedding matrix modulus on the tissue mechanical maps. Maybe cell viability after could be quantified for the different matrix composition and modulus and slicing protocol. Also is there any link between the cell viability and measured mechanical properties?

For all AFM images – always indicate with which kind of probes have been obtained. Add scale bars – e.g. Fig 5. Discuss statistics of the AFM data.

Author Response

Reviewer 2.

Thank you very much for your thoughtful criticism, that helped greatly to improve our manuscript.

Your recommendations were accepted by us. We provided additional experiment and added new data. Below please find our responses carefully addressed to your comments.

  1. Reviewer 2 wrote:

“This paper describes a method to embed brain tissue samples for AFM imaging of morphological features. The authors argue that currently no method exists for living tissue AFM imaging. However, to our knowledge AFM living tissue imaging, even in vivo – both in terms of mechanical properties (i.e., modulus) and topography – has been performed multiple times. The advantages and the influence of the preparation method on the extracted tissue properties are not discussed or not quantitatively assessed. The methodology is not discussed in detail (e.g. AFM probe specifications and imaging parameters for all the discussed imaging modes are not given, also no info on the number of samples and data statistics). Cell viability is not quantified. Scale bars and imaging specs are not given for most of the figures. The language is sometimes confusing. The manuscript should be re-written.

See for instance

Krieg, M., Fläschner, G., Alsteens, D. et al. Atomic force microscopy-based mechanobiology. Nat Rev Phys 1, 41–57 (2019).  

https://www.ncbi.nlm.nih.gov/pmc/articles/PMC6587663/

The main novelty in respect to state of the art should be better discussed in the introduction session. This part should be re-written as it is unclear – some general statements are made without sufficient arguments to make a point. The language is sometimes confusing and should be improved. Specifically, the proposed technique should be introduced in the context of the literature relevant for tissue embedding for AFM to clarify novelty and advances in respect to state of the art. The choice of the embedding materials should be justified.

Author’s response:

We want to draw your attention to the fact that our work is focused on a new method of combined morphological and mechanical mapping of live brain tissue (lines 442-445). Indeed, many live tissues have already been mapped on AFM. As for the nervous tissue, as an example of extremely soft one, a comprehensive method for studying it with the help of AFM has not yet been presented in the literature. All works of similar subjects, including cited by you, contain shortcomings.

  1. In this work [https://www.ncbi.nlm.nih.gov/pmc/articles/PMC6587663/], Figure 1 contains a mechanical and morphological map of the surface area of decellularized dermal matrix. As can be seen from the mechanical map, such a matrix has a sufficiently rigid structure, with a maximum elasticity index of up to 30 kPa. The more rigid and elastic the surface is, the easier it is to study it with AFM. However, the morphological map contains artifacts, and the size of the field of view is only 20 by 20 microns. If in this particular case, when the cells were removed from the matrix, such a size of the area is acceptable, then in the case of the analysis of the whole tissue, it is much more correct to study a larger area.
  2. In this work [https://www.ncbi.nlm.nih.gov/pmc/articles/PMC6587663/], Figure 2 shows a cross section of the spinal cord, as well as a mechanical map showing differences in the rheology of white and gray matter. Such a map can be classified as macrorheological, since it does not allow to evaluate the mechanics at the micron level.
  3. A number of works describe the study of fixed and cryotomed tissue, but it has been proven that their mechanics change significantly, not to mention damage to the structure at the micro level [https://aip.scitation.org/doi/abs/10.1063/1.5001579?journalCode=ap].
  4. In the paper [https://link.springer.com/article/10.1007/s00429-017-1486-z], a topographic map of the nervous tissue is presented, but there is no mechanical mapping, while it is precisely the correlation between microarchitectonics and nanomechanics that is of greatest interest.

Nevertheless, we made the necessary changes to the introduction and discussion, where we cited relevant works and discussed their shortcomings (lines 67-75).

  1. Reviewer 2 wrote:

“Concerning the methodology  - section 2.5. AFM – all procedural details should be given – e.g. how many samples have been analyzed for each condition, fitting formula’s used for modulus and stiffness calculation. For all AFM images – always indicate with which kind of probes have been obtained. Add scale bars – e.g. Fig 5. Discuss statistics of the AFM data.”

Author’s response:

We have added a detailed description of all the probes used, scanning modes, named the number of samples studied, provided statistical evidence and fitting formula’s used for modulus and stiffness calculation. We rewrite AFM protocol (section 2.5, lines 178-211)

  1. Reviewer 2 wrote:

“Modulus maps are not clear in the manuscript – it would make sense maybe to include modulus histograms to quantify the effect of the different sample preparation conditions. It would be interesting to check the effect of the embedding matrix modulus on the tissue mechanical maps. Maybe cell viability after could be quantified for the different matrix composition and modulus and slicing protocol. Also is there any link between the cell viability and measured mechanical properties?”

Author’s response:

We performed an additional experiment and we proved that our slices preparation technique for AFM, using embedding matrices of different compositions, does not change the rheological properties of tissues and their viability. You can see the results in Figure 1, which we added to the manuscript. Thus, we have demonstrated that, on the one hand, our method does not affect the parameters under study (stiffness, elasticity, viability), and, on the other hand, it will allow obtaining high-quality morphological and nanomechanical maps of nervous live tissue. We added protocol to lines 270-275.

Cell viability in tissue sections was quantified using the Imaris software, statistical results are presented at figure 1.

We carried out a complete revision of the manuscript, changed the captions to the figures, making them more accurate and understandable, added scale bars, indicated the types of probes with which maps and data were obtained.

An important step in the preparation of slices suitable for AFM is vibratomy. You recommended us to study - how different modes of vibratomy affect the viability of the slices. The vibratomy mode described in our work (blade movement speed, vibration frequency) (lines 145-150) is the only satisfactory one for the task of obtaining slices with appropriate quality. We used maximum vibration frequency (50 Hz). With a decrease in the vibration frequency, a visible deterioration in the quality of the cut occurs, expressed in its raggedness. We used the minimum blade speed (0.2mm/sec). With an increase in the speed of the blade, the tissue compresses and breaks into tiny particles. As evidence, we provide video of the vibratomy process.

  1. Reviewer 2 wrote:

“Most relevant references should be cited.”

Author’s response:  We cited more modern researches of tissue embedding and live tissue AFM, added 10 references.

Reviewer 3 Report

Farniev et al., submitted the paper entitled “Nanomechanical and morphological AFM mapping of normal tissues and tumors on live brain slices using specially designed embedding matrix and laser-shaped cantilevers”, to publish in “Biomedicines (I.F = 6.081)”. This work is an impressive one, can be accepted after addressing the queries.

  1. In few sections, results and discussion are seems to be deficient, which should be upgraded. Especially discussion for Figures 2-5 requires enrichment.
  2. Improve the resolution of insets 3D images of Figures 3b,c and 4b,c.
  3. Provide scale bars for Figure 5.
  4. Conclusion section must be revised by providing merits, limitations, conditions and future directions.

Author Response

We express our deepest gratitude for your appreciation of our manuscript. We have made the necessary changes according to your comments. Below we have listed them:

  1. Reviewer 3 wrote

“In few sections, results and discussion are seems to be deficient, which should be upgraded. Especially discussion for Figures 2-5 requires enrichment.”

Author’s response:

We have improved the description of the results and the discussion for Figures 3, 4, 5 and 6 (lines 539-553, 398-400, 379-381, 350-353).

  1. Reviewer 3 wrote

“Improve the resolution of insets 3D images of Figures 3b,c and 4b,c.”

Author’s response:

We improved resolution of 3D insets.

  1. Reviewer 3 wrote

“Provide scale bars for Figure 5.”

Author’s response:

Figure 5 now is named figure 6, we added scale bar for it.

  1. Reviewer 3 wrote

Conclusion section must be revised by providing merits, limitations, conditions and future directions.

Author’s response

We have added the conclusion section, considering your comments. (lines 588-613)